# *Solidago graminifolia* L. Salisb. (*Asteraceae*) as a Valuable Source of Bioactive Polyphenols: HPLC Profile, In Vitro Antioxidant and Antimicrobial Potential

**DOI:** 10.3390/molecules24142666

**Published:** 2019-07-23

**Authors:** Anca Toiu, Laurian Vlase, Dan Cristian Vodnar, Ana-Maria Gheldiu, Ilioara Oniga

**Affiliations:** 1Department of Pharmacognosy, “Iuliu Hațieganu” University of Medicine and Pharmacy, 400012 Cluj-Napoca, Romania; 2Department of Pharmaceutical Technology and Biopharmacy, “Iuliu Hațieganu” University of Medicine and Pharmacy, 400012 Cluj-Napoca, Romania; 3Department of Food Science, Faculty of Food Science and Technology, University of Agricultural Sciences and Veterinary Medicine, 400372 Cluj-Napoca, Romania; 4Department of Pharmaceutical Botany, “Iuliu Hațieganu” University of Medicine and Pharmacy, 400012 Cluj-Napoca, Romania

**Keywords:** *Solidago graminifolia*, polyphenolic compounds, antioxidant, antimicrobial

## Abstract

*Solidago* species are often used in traditional medicine as anti-inflammatory, diuretic, wound-healing and antimicrobial agents. Still, the bioactive compounds and biological activities of some species have not been studied. The present work aimed to investigate the polyphenolic profile and the biological properties of *Solidago graminifolia* L. Salisb., a poorly explored medicinal plant. The hydroalcoholic extracts from aerial parts were evaluated for total phenolic content (TPC), total flavonoid content (TFC) and the polyphenolic compounds were investigated by HPLC-MS. The antioxidant potential in vitro was determined using DPPH and FRAP assays. Antibacterial and antifungal effects were evaluated by dilution assays and MIC, MBC and MFC were calculated. The results showed that *Solidago graminifolia* aerial parts contain an important amount of total phenolics (192.69 mg GAE/g) and flavonoids (151.41 mg RE/g), with chlorogenic acid and quercitrin as major constituents. The hydroalcoholic extracts showed promising antioxidant and antimicrobial potential, with potent antibacterial activity against *Staphylococcus aureus* and important antifungal effect against *Candida albicans* and *C. parapsilosis*. The obtained results indicated that the aerial parts of *Solidago graminifolia* could be used as novel resource of phytochemicals in herbal preparations with antioxidant and antimicrobial activities.

## 1. Introduction

The genus *Solidago* (Asteraceae family) contains about 130 species widespread throughout the world. Most are herbaceous flowering plants which grow wild or are cultivated especially for ornamental purposes. Several species have been used in traditional medicine in many continents: *Solidago virgaurea* L. (European Goldenrod) in Europe and Asia, *S. canadensis* L. (Canadian Goldenrod)*, S. gigantea* Aiton (Giant Goldenrod) and *S. odora* Aiton in North America, *S. chilensis* Meyen in South America [1,2].

Various bioactive compounds such as flavonoids, caffeoylquinic acid derivatives, salicylic acid derivatives, triterpenoid saponines, clerodane-type diterpenes, polysaccharides, and essential oils were previously reported in *Solidago* species [3,4,5,6]. Previous studies have demonstrated that these natural products from *Solidago* species possess a broad spectrum of biological activities: antioxidant [1,7,8], antimicrobial [1,5,7,9], spasmolytic [10], anti-inflammatory [2,3,10,11], anti-obesity [12,13], cytotoxic [14,15], gastroprotective effects [16,17,18]. The diuretic activity was attributed to flavonoids and leiocarposide [19,20,21], while the anti-inflammatory effect to phenolic compounds (quercitrin, hyperoside, leiocarposide) from *Solidago* species extracts [2,3,11,22,23,24,25]. As several studies suggested, the cytotoxic activity may be due to terpenic compounds present in *Solidago* sp. extracts: triterpenes, diterpenes and sesquiterpenes [26,27,28].

*S. virgaurea* L. is considered one of the most important species of the *Solidago* genus. The aerial parts have been used for centuries as anti-inflammatory, spasmolytic and diuretic agents in traditional medicine for the treatment of numerous symptoms, especially as a urological remedy in kidney and bladder inflammation, urolithiasis, cystitis [1,3,29,30]. *S. virgaurea* extracts contain flavonoids (mostly quercetin glycosides), salycilic acid derivatives (leiocarposide, virgaureoside), caffeoylquinic acid derivatives (chlorogenic acid, caffeic acid), triterpene saponins (oleanane type), tannins, essential oils. Several bioactive compounds from *S. virgaurea* extracts have synergic action: flavonoids, saponins, caffeic acid derivatives and leiocarposide displayed anti-inflammatory activity [24,31], polyphenolic compounds showed antioxidant effects [7,32,33], while flavonoids determined the spasmolytic effects [34,35,36]. In vivo and in vitro studies sustain some other biological activities: antibacterial [5,7,9,37,38], antifungal [28,39], anticancer [14,27,28], immunomodulatory [27,40], diuretic [20,21], cardioprotective [41].

According to European Pharmacopoeia, flavonoids are quality marker compounds: the vegetal product *Solidaginis herba* should contain at least 2.5% flavonoids (expressed as hyperoside), while in *Solidaginis virgaureae herba* their concentration should be between 0.5 and 1.5%.

*S. canadensis* and *S. gigantea* are species native to North America and naturalized in Europe, where were introduced as ornamental. Afterwards, they became invasive in several countries and are regarded as problematic, since they might have a negative ecological impact [1]. Still, North American Goldenrods (*S. canadensis* and *S. gigantea*) are now recognized as important medicinal plants and are used in phytotherapy taking into account that their aerial parts (*Solidaginis herba*) have a higher content of flavonoids and saponins then *Solidaginis virgaureae herba* [1].

*Solidago graminifolia* L. Salisb. (syn. *Euthamia graminifolia* (L.) Nutt) is an herbaceous perennial plant, with linear-lanceolate leaves, persistent rhizomes and small yellow flowers in corymbose panicles [42,43]. Although it was also naturalized from North America and brought at almost the same time as *S. canadensis* and *S. gigantea*, *S. graminifolia* remained restricted to isolated regions in Europe [6,42].

Few studies on *S. graminifolia* bioactive secondary metabolites showed that roots contain acetylenes [44], while two di-C-glycosylflavones (schaftoside, isoschaftoside) [45] and essential oil (1.7%) [46] were isolated from aerial parts. Phenolic compounds [47] and essential oil were produced by *Solidago graminifolia* in vitro cultures and small quantitative differences were detected between the micropropagated and the wild plants [6]. Previous research showed that *S. graminifolia* methanol extract may be used as natural amoebicidal agent [48]. To the best of our knowledge, there are no reports yet on antioxidant, antibacterial and antifungal activities of *S. graminifolia* aerial parts. Hence, the aim of the present study was to investigate the polyphenolic profile and some biological properties of *Solidago graminifolia* aerial parts extracts.

## 2. Results and Discussion

### 2.1. The Quantification of Total Bioactive Compounds

Several in vitro and in vivo studies revealed that the presence of polyphenolic compounds in plant extracts could be related with important biological properties, such as antioxidant, immunomodulatory, antimicrobial, anticancer, prebiotic-like, vasodilating activities [49,50]. Additionally, their preventive effects on chronic diseases, in particular neurodegenerative, cardiovascular, obesity, osteoporosis or type 2 diabetes were confirmed by many epidemiological studies [51,52,53].

Since flavonoids and phenolic acids are among the most important phytochemical constituents of *Solidago* species, we evaluated their content in several *S. graminifolia* extracts. The results of the quantitative determination of total phenolics and flavonoids from different extracts, as well as the yield of crude extracts (mg extract/g plant material) are presented in Table 1. The extraction of bioactive compounds from *S. graminifolia* aerial parts with polar solvents (aqueous extract, methanol and ethanol extracts) yielded 280.01 mg/g, 295.4 mg/g and 317.17 mg/g respectively, which is much higher than that of non-polar solvents (chloroform extract, petroleum ether extract) which yielded 40.5 mg/g and 121.2 mg/g respectively.

Total phenolic contents (TPC) and total flavonoid contents (TFC) of the extracts were determined; the highest values were observed in the *S. graminifolia* ethanol extract, with total phenolics (192.69 mg GAE/g extract) and flavonoids (151.41 mg RE/g extract). The TPC and TFC for methanol extract (179.04 mg GAE/g, 130.58 RE/g) and aqueous extract (166.29 mg GAE/g, 128.37 RE/g) were found to be similar. The lower TPC and TFC values were determined in chloroform extract, while in petroleum ether extract the amounts were almost two times higher (40.67 mg GAE/g, 98.44 RE/g).

The obtained results were in good agreement with the ones presented by Kołodziej et al. [9], who determined the dried matter, *o*-dihydroxyphenols and flavonoid contents in hexane and ethanol extracts from *S. virgaurea*, *S. canadensis* and *S. gigantea*. Similarly, the highest contents of bioactive compounds were found in ethanol extracts (118 mg/g, 156 mg/g and 156 mg/g dry matter) [9], which are lower than those from *S. graminifolia* extract in this report. The higher content in total polyphenols and flavonoids from *S. graminifolia* aerial parts ethanol extract might be due to selecting optimal extraction conditions: 50 min extraction time, 60 °C temperature, and 1:20 the ratio of sample to 70% (*v*/*v*) ethanol. The extraction conditions used in the present study are in agreement with other research on increasing the number of phenolic compounds by proper selection of extraction condition [54]. Several studies pointed out that certain factors, such as the extraction method, the type of solvents and solubility of active compounds, the temperature, the extraction time, and the ratio of solvent-to-sample are essential parameters which greatly influence the yield and quality of obtained extracts from plant materials [55,56,57]. Recent research carried out by Woźniak et al. [1] on *Solidago* sp. showed that the methanol extracts from aerial parts contain 146.1 mg flavonoids/g extract and 170.5 mg polyphenols/g extract for *S. gigantea*, and 82.1 mg flavonoids/g extract and 130.5 mg polyphenols/g extract for *S. canadensis*, which is lower than the content determined in *Solidago* species considered in our study.

Also, it should be noted that the flavonoid content meets the quality criteria of European Pharmacopoeia mentioned at *Solidaginis herba* (minimum 2.5% flavonoids).

Our study showed that *S. graminifolia* ethanol extract contains important amounts of phenolic compounds and flavonoids, which could be better exploited as valuable source of phytochemicals in herbal remedies.

### 2.2. Qualitative and Quantitative Analysis of Polyphenols

Taking into account that the highest amounts of total flavonoids and total phenolics were observed in *S. graminifolia* ethanol extract, the HPLC analysis for identification and quantification of polyphenolic compounds was carried out and the obtained results are presented in Table 2.

Using sensitive, precise and simple analytical HPLC methods, the presence of flavonoid glycosides (hyperoside, rutin, quercitrin), flavonoid aglycones (quercetin, luteolin, kaempferol), and phenolic acids (caftaric, gentisic, chlorogenic, *p*-coumaric, ferulic, gallic, protocatechuic, vanillic, syringic, rosmarinic acids) was detected in *Solidago graminifolia* aerial parts. The investigation showed that chlorogenic acid (997.88 mg/100 g extract), quercitrin (431.59 mg/100 g) and hyperoside (253.19 mg/100 g) were the major active compounds in *S. graminifolia* aerial parts extract. The chromatogram of *Solidago graminifolia* ethanol extract using the HPLC method *b*, as described in Materials and methods Section 3.5 is presented in Figure 1.

Protocatechuic acid was assessed as >60 mg/100 g extract, while rosmarinic acid was assessed as >10 mg/100 g extract. Due to their low amount (<LOQ), three compounds (caftaric acid, gentisic acid, rutin) were only identified in the extract.

The information on the polyphenolic compounds from *S. graminifolia* aerial parts is scarce. Previous studies identified two C-glycosylflavones (schaftoside, isoschaftoside) in naturally growing plants and the ones from in vitro culture [45,47].

Chlorogenic acid (5-*O*-caffeoylquinic acid) is a plant secondary metabolite widely distributed in coffee, tea, many fruits, vegetables and herbs. Recent studies on this phenolic compound have demonstrated multiple biological properties, such as antioxidant, anti-inflammatory, antibacterial, antiviral, cardioprotective, hepatoprotective, neuroprotective, antihypertensive, anti-obesity, antidiabetic, antiapoptotic [58,59]. Additionally, many in vivo studies and clinical trials have been done concerning the health benefits of chlorogenic acid as a nutraceutical agent for prevention and treatment of metabolic syndrome and associated diseases [60].

Quercitrin (quercetin 3-rhamnoside) was the main flavonoid identified in *S. graminifolia* aerial parts extract. This compound is the precursor of quercetin, which has been previously described as an important contributor to the antioxidant, anti-inflammatory, vasodilator, antiatherosclerotic, antihypercholesterolemic, anti-obesity and angioprotective potential of plant extracts [61,62].

Due to their multiple biological activities, the phenolic compounds from several *Solidago* species have been formerly examined. *S. gigantea* and *S. canadensis* methanol extracts contain lower amounts of chlorogenic acid (0.1 mg/g dw plant material) while higher content of quercitrin was determined in *S. gigantea* [1] compared with *S. graminifolia* from our study. Another research on *S. canadensis* hydroalcoholic extract found rutin (8.93% *w*/*w*) as the main compound, as well as caffeic acid derivatives [25].

Because the most significant differences between the *Solidago* species were determined by quercitrin, rutin and chlorogenic acids, establishing the metabolic profile of polyphenols may contribute to ensure plants material of high quality and safety for more efficient phytopharmaceuticals.

Considering that plant extracts are complex mixtures of numerous natural compounds with synergic or additive effects, the investigation on new sources of chlorogenic acid could reveal possible applications in medicine and pharmacy.

### 2.3. The Evaluation of In Vitro Antioxidant Activity

Since higher amounts of polyphenols were accounted in *S. graminifolia* polar extracts, the evaluation of the biological activities was carried out and the obtained results are presented in Table 3.

The antioxidant activity evaluation was carried out by DPPH (2,2-diphenyl-1-picrylhydrazyl) and ABTS (2,2′-azinobis-(3-ethylbenzothiazoline-6-sulfonic acid)) radical scavenging activity assays and in relation to the reference antioxidant Trolox (IC_50_ = 11.2 ± 0.21 µg/mL). These frequently used methods are rapid and valuable for the evaluation and quantification of the free radical scavenging activity of natural compounds from plant extracts.

The high content of total polyphenols contributed to important antioxidant capacity in regard to free radical scavenging effect. The reduced IC_50_ values for *S. graminifolia* aerial parts extracts requested smaller sample concentration to reveal 50% of the inhibitory effect. The evaluation of in vitro antioxidant activity of *S. graminifolia* aerial parts extracts was performed for the first time and the IC_50_ values were calculated. The highest scavenging activity was detected for *S. graminifolia* ethanol extract (IC_50_ = 12.61 μg/mL), which is the richest in phenolic compounds, and the results are comparable with the ones obtained for the reference compound Trolox. The results of our research indicate that the antioxidant potential may be due to the presence of phenolic acids and flavonoids, which were found in significant amounts in *S. graminifolia* ethanol extract. Furthermore, the good radical scavenging activity could indicate that the extract might act as a primary antioxidant. Several research reported the strong correlation between the phenolic content and radical scavenging activity [63,64].

Each phenolic compound react differently and its antioxidant effect is closely related with the structure, for example quercetin presents iron-chelating and iron-stabilizing effects, while chlorogenic acid inhibit lipid oxidation in oil-in-water emulsion through a complex effect, metal-chelating in the hydrophilic phase and free radical scavenging in the hydrophobic phase [60,65].

Recent research on *S. chilensis* leaves indicated that the methanol extract contain quercitrin and afzelin with gastric healing effects related to antioxidant and antisecretory properties, as well as the beneficial effects on the mucus protection [16]. Furthermore, the scavenger activity of quercitrin and the extract was showed in the DPPH assay. Therefore, the antioxidant potential was reported as a complex process mediated by an increased level of the endogen antioxidants, as well as by a direct action as a free radical scavenger [16]. Also, the in vitro antioxidant properties of quercitrin are well documented by several studies [66,67].

Our preliminary investigation on in vitro antioxidant activity of *S. graminifolia* aerial parts revealed important free-radical scavenging effects that need to be confirmed by further in vivo studies, as well as the underlying mechanisms of these properties.

### 2.4. The Evaluation of Antibacterial Activity

The evaluation of antibacterial potential of *S. graminifolia* extracts was assessed by microdilution assay, gentamycin as standard antibiotic and different Gram+ and Gram- bacteria (*Staphylococcus aureus*, *Listeria monocytogenes*, *Pseudomonas aeruginosa*, *Salmonella typhimurium* and *Escherichia coli*). Minimum Bactericidal Concentration (MBC) and Minimum Inhibitory Concentration (MIC) were determined and the results are presented in Table 4. The ethanol extract had MIC levels between 0.048–3.12 mg/mL, while for the methanol and the aqueous extracts they were between 0.096–3.12 mg/mL. The best antibacterial effect was observed for *S. graminifolia* ethanol extract against *S. aureus* (MIC = 0.048 mg/mL and MBC = 0.096), whilst similar activities were noted against *P. aeruginosa*, *L. monocytogenes* and *Escherichia coli*. All extracts had the lowest effects against *Salmonella typhimurium*. As reported by Salvat et al., MIC amounts around or less than 0.5 mg/mL suggest good antibacterial activity [68], hence the *S. graminifolia* ethanol extract showed significant antimicrobial potential against *S. aureus* and moderate versus the other four bacterial strains. Overall, the most effective extract was *S. graminifolia* ethanol extract. Several studies demonstrated significant antibacterial activity exerted by polyphenolic compounds [69,70]. Taking into account the results obtained for phytochemical characterization of *S. graminifolia* extracts, it can be stated that the higher amounts of polyphenols from ethanol extract might determine better antibacterial potential. Furthermore, chlorogenic acid proved important antimicrobial activity against several bacterial strains (*Klebsiella pneumoniae*, *Escherichia coli*, *S. aureus*, *P. aeruginosa* [71], *H. pylori* [72]. Accordingly, it could be used as preservative in pharmaceutical or cosmetic products, as well as a food additive [58].

Few studies evaluated the antimicrobial activity of several *Solidago* species. The ethanol and hexane extracts from *S. virgaurea*, *S. gigantea* and *S. canadensis* aerial parts presented antibacterial effect and acted strongly to Gram + (*B. subtilis*, *S. aureus*, *S. faecalis*) than Gram − bacteria (*K. pneumoniae*, *E. coli*, *P. aeruginosa*). The best effect was observed for *S. canadensis* hexane extract (MIC values from 5 to 10 mg/mL against Gram + bacteria) and the ethanol extracts from *S. gigantea* and *S. canadensis* (MIC values from 10 to 50 mg/mL against Gram + bacteria) [9]. The results were in agreement with the antibacterial activity of *S. virgaurea* against similar bacterial strains from other studies [7,38].

The essential oil from *S. canadensis* roots showed significant antibacterial effect against *E. coli* and *S. faecalis* and moderate antifungal activity against *C. albicans* [73]. The essential oil from *S. microglossa* aerial parts exhibited concentration-dependent activity against seven bacterial strains and two yeasts, while the methanol extract showed mild activity (MIC > 1 mg/mL) [74].

### 2.5. The evaluation of Antifungal Activity

The evaluation of antifungal activity of *S. graminifolia* extracts was performed using five species of fungi, chosen based on their relevance for public health. The data obtained for the assessement of antifungal potential (Minimum Fungicidal Concentration, MFC, and Minimum Inhibitory Concentration, MIC) are presented in Table 5.

*Candida albicans* and *C. parapsilosis* showed the highest sensitivity (MIC value 0.012 mg/mL and MFC value 0.025 mg/mL), while *Aspergillus flavus*, *A. niger* and *Penicillium fumiculosum* presented similar sensitivity (MIC value 0.05 mg/mL and MFC value 0.1 mg/mL) to *S. graminifolia* aerial parts ethanol extract.

The most resistant strains were *Aspergillus flavus*, *A. niger* and *Penicillium fumiculosum* against water extract, with MIC value of 0.1 mg/mL.

Generally, the most effective extract against all species of tested fungi was *S. graminifolia* ethanol extract. The results are in accordance with previous research which provide substantial evidence that many alcohol extracts possess antifungal activity considering that less polar compounds occur and a more complete extraction is performed with alcohol [75,76]. Moreover, former studies revealed the antifungal activity of chlorogenic acid [77,78], the main compound from *S. graminifolia* ethanol extract. The phenolic compound presented significant in vitro antifungal effect against *C. albicans*, *M. furfur* and *T. beigelii* by disrupting the structure of the cell membrane (MIC values between 40–80 µg/mL) [79].

The antifungal potential of some *Solidago* species has been previously evaluated. *S. gigantea* aqueous extract from flowers presented a significant degree of antifungal activity towards most of the yeast isolates [75]. The bioassay-guided phytochemical investigation on *S. virgaurea alpestris* performed by Laurençon et al. [4] showed that the isolated triterpenoid saponins determined significant inhibition of *C. albicans* yeast-hyphal conversion. Accordingly, the extract might preserve the native oral biofilm while protecting the patiens from virulent *C. albicans* strains [4].

The investigation on the antifungal activity of ethanol extracts from the aerial parts showed the efficacy of *S. virgaurea* against some dermatophytes (*Microsporum canis*, *Microsporum gypseum*, and *Trichophyton mentagrophytes*), while *S. gigantea* was active against *Candida pseudotropicalis* [80]. Also, the essential oil isolated from *S. chilensis* leaves was efficient against the dermatophytes *T. mentagrophytes* and *M. gypseum* [81].

## 3. Materials and Methods

### 3.1. Chemicals and Reagents

For the phytochemical analysis by LC-MS/MS, a number of 25 polyphenols were used as standards. The following standards were purchased from Sigma (Sigma-Aldrich Chemie GmbH, Schnelldorf, Germany): caftaric acid (≥97%), chlorogenic acid (≥95%), caffeic acid (≥98%), *p*-coumaric acid (≥98%), rutin (≥94%), apigenin (≥95%), kaempferol (≥97%), luteolin (≥98%), gentisic acid (≥98%), myricetol (≥96%), fisetin (≥98%), (+)-catechin (≥96%), (−)-epicatechin (≥90%), quercetin (≥95%), quercitrin (quercetin 3-rhamnoside) (≥78%), isoquercitrin (quercetin 3-d-glucoside) (≥98%), hyperoside (quercetin 3-d-galactoside) (≥97%), patuletin (≥98%), protocatechuic acid (3,4-dihydroxybenzoic acid) (≥97%), syringic acid (≥95%), vanillic acid (≥97%), rosmarinic acid (≥96%). Gallic acid (≥98%) was acquired from Merck (Darmstadt, Germany). Ferulic acid (≥99%) and sinapic acid (≥98%) were obtained from Roth (Karlsruhe, Germany). Solvents used for extraction and separation were of HPLC analytical-grade (methanol, ammonium acetate, acetonitrile) or analytical-grade (petroleum ether, chloroform, hydrochloric acid, acetic acid, potassium hydroxide). These solvents and Folin-Ciocâlteu reagent were purchased from Merck (Darmstadt, Germany). 6-hydroxy-2,5,7,8-tetramethylchroman-2-carboxylic acid (Trolox) and 2,2-diphenyl-1-picrylhydrazyl (DPPH) were purchased from Alfa-Aesar (Karlsruhe, Germany).

### 3.2. Plant Material

*Solidago graminifolia* aerial parts were harvested at full flowering stage from Cluj County in July 2016. The plant material from Romanian spontaneous flora was obtained and authenticated by Prof. M. Tămaş from the Department of Pharmaceutical Botany. A voucher specimen was deposited in the Herbarium of Pharmacognosy Department, University of Medicine and Pharmacy, Cluj-Napoca (SG-2).

### 3.3. Preparation of Natural Extracts

The extraction of vegetal product was performed with 70% methanol, 70% ethanol, water, petroleum ether, chloroform, for 50 min at 60 °C and 1:20 ratio of sample to solvent (*v*/*v*) [55]. The extracts were concentrated to dryness under reduced pressure at 40 °C. The dried extracts were stored at 4 °C until further analysis. The extracts yields were expressed in relation to dry weight of vegetal product (mg crude extract/g dry weight plant material).

### 3.4. Quantitative Analyses

The total phenolic content (TPC) of the extracts from *Solidago graminifolia* aerial parts was determined by Folin-Ciocâlteu method as detailed in previous paper [82]. The content in total phenolics was expressed as mg gallic acid equivalents (GAEs)/g dry weight (dw) vegetal product. The experiments were performed in triplicate.

The evaluation of total flavonoid content (TFC) of the extracts from *Solidago graminifolia* aerial parts was performed by using a spectrophotometric method [83]. The flavonoids content was expressed as rutin equivalents (mg REs)/g dw vegetal product.

### 3.5. LC-MS/MS Analysis of Polyphenols. Apparatus and Chromatographic Conditions

The phytochemical profile of *Solidago graminifolia* ethanol extract was assessed by liquid chromatography- tandem mass spectrometry (LC-MS/MS). The equipment was an Agilent 1100 HPLC Series system (Agilent, Santa Clara, CA, USA) equipped with auto sampler, binary gradient pump, degasser, column thermostat (set at 48 °C), and UV detector. The mass spectrometer was an Agilent Ion Trap 1100 SL (LC/MSD Ion Trap VL, Agilent, Santa Clara, CA, USA) equipped with electrospray ionization (ESI) or atmospheric pressure chemical ionization (APCI). The flow rate was 1 mL/min and the injection volume was 5 µL. The separation of compounds was performed on a reverse-phase analytical column (Zorbax SB-C18 100 × 3.0 mm i.d., 3.5 μm particle).

For identification and quantification of rosmarinic acid, the mobile phase was a mixture of acetonitrile and ammonium acetate in water (1 mM). The elution was in gradient, starting with 5% acetonitrile up to 25% acetonitrile at 3.3 min. The MS with ESI source operated in negative ionization mode, set for fragmentation and isolation of deprotonated rosmarinic acid molecule with *m*/*z* = 359, and the quantification of this compound was performed based on the deprotonated molecule (method a).

For the other 24 polyphenolic compounds, two distinct analytical methods were used. One method was applied for the identification of the following 18 polyphenols: caftaric acid, gentisic acid, caffeic acid, chlorogenic acid, *p*-coumaric acid, ferulic acid, sinapic acid, hyperoside, isoquercitrin, rutin, myricetol, fisetin, quercitrin, quercetin, patuletin, luteolin, kaempferol, and apigenin. In this case, the mobile phase was a mixture of methanol: 0.1% acetic acid (*v*/*v*). The elution gradient was binary, starting with a linear gradient (5% methanol at start, 42% methanol at 35 min) and for the next 3 min followed by an isocratic elution (42% methanol). Both UV and MS mode were used for detection of the compounds. For detection of polyphenolic acids, the UV detector was set at 330 nm during the first 17 min, while for detection of flavonoids and their aglycones, the wavelength was switched to 370 nm until 38 min. For the MS system, the operating condition was in negative ionization mode using an electrospray ion source (capillary voltage +3000 V, nebulizer 60 psi (nitrogen), dry gas nitrogen at 12 L/min, dry gas temperature 360 °C). For the identification of the previously mentioned 18 polyphenols, the mass spectra signal specific for each compound was used, whereas for their quantification, the UV trace assisted by MS was employed. A standard solution of polyphenols was used for collecting all spectra and integrating them in a library. The minimal concentration which produced a reproductive peak characterized by a signal-to-noise ratio greater than three was considered for calculation of the detection limits (method b).

Another LC-MS method was employed for the following 6 polyphenols: epicatechin, catechin, syringic acid, gallic acid, protocatechuic acid, and vanillic acid. The working conditions for this method were similar with those aforementioned apart from the mobile phase binary gradient, which was 3% methanol at start, 8% methanol at 3 min, and 20% methanol from 8.5 min to 10 min. For these 6 polyphenols, both identification and quantification were carried out on MS mode, for which the working conditions were previously described (method c).

DataAnalysis and ChemStation software (Agilent Inc., Santa Clara, CA, USA) were used for chromatographic data collection and processing.

The retention times for the compounds were determined using reference standards and were based on the mass spectrum for each compound. Spiking samples with a solution containing each polyphenol (10 µg/mL) was used for accuracy check.

For identification of compounds, their retention times and the recorded ESI-MS spectra were compared with those of standards, which were obtained under identical working conditions. The method of external standard was employed for the quantification of polyphenols in each extract and the calibration curves for a five point plot were linear in the range 0.5–50 µg/mL (R^2^ > 0.999).

### 3.6. The Evaluation of In Vitro Antioxidant Activity

The determination of the radical scavenging effect of *Solidago graminifolia* extracts was achieved by DPPH and ABTS radical scavenging assays and detailed in previous paper [84]. Trolox was used as reference standard. For DPPH assay the antioxidant potential was expressed as IC_50_ (μg/mL), while for ABTS assay the results were expressed as Trolox equivalents (TEs)/g of extract.

### 3.7. The Evaluation of Antibacterial Activity

For the assessment of antibacterial potential, two Gram-positive bacteria (*Staphylococcus aureus*, ATCC 49444 and *Listeria monocytogenes*, ATCC 19114) and three Gram-negative bacteria (*Escherichia coli*, ATCC 25922, *Pseudomonas aeruginosa,* ATCC 27853, *Salmonella typhimurium*, ATCC 14028) were provided by Food Biotechnology Laboratory, USAMV Cluj Napoca. The experiments were carried out using Muller-Hinton Agar, at 4 °C.

In order to evaluate the antibacterial potential of plant extracts, an adjusted microdilution method was employed [83,84]. The determination of minimum inhibitory concentrations (MICs) was done by dilution method, using 96 multi-well plates. Several extract dilutions from *Solidago graminifolia* aerial parts were blended with 10 μL of inoculum and 100 μL of Tryptic Soy Broth, and then were incubated at 37 °C, for 24–48 h. MIC represented the smallest drug concentration which could prevent the change of colour, while the minimum bactericidal concentration (MBC) represents the smallest concentration which indicates 99.5% killing of the original inoculums. The positive control was the standard antibiotic Gentamycin (25 μL/well, 4 μg/mL), while 70% ethanol represented the negative control. The analyses were carried out three times, then the averages were calculated [83,84].

### 3.8. The Evaluation of Antifungal Activity

In order to determine the antifungal potential of *Solidago graminifolia*, five fungal strains: *Aspergillus niger* ATCC 6275, *Aspergillus flavus* ATCC 9643, *Penicillium funiculosum* ATCC 56755, *Candida parapsilosis* ATCC 22019, *Candida albicans* ATCC 10231 obtained from Food Biotechnology Laboratory (USAMV Cluj Napoca) were used. We employed a microdilution method detailed previously and we determined the minimum inhibitory concentration (MIC) and minimum fungicidal concentration (MFC). The experiments were done in duplicate and repeated three times [83,84].

## 4. Conclusions

*S. graminifolia* is a less studied species of the genus *Solidago*, which includes medicinal plants known for their diuretic, anti-inflammatory, antimicrobial, antioxidant, spasmolytic properties. The active compounds are saponins, flavonoids, salicylic acid derivatives, tannins, etc. The aim of our study was to analyse the chemical compounds and some biological properties of *S. graminifolia*, in order to evaluate its therapeutic potential. In this way, we studied the content of phenolic compounds in different extracts and the best results were obtained for the ethanol extract. Several phenolic compounds were identified by HPLC in *S. graminifolia* aerial parts ethanol extract, with important amounts of chlorogenic acid and quercitrin. Qualitative and quantitative differences comparing to other *Solidago* species were emphasised. The high total phenolic content might be related with the good antioxidant and antimicrobial activities revealed for the hydroalcoholic extract. An important antibacterial effect against *Staphyloccus aureus* and a potent antifungal activity against *Candida albicans* and *C. parapsilosis* were noted.

Our preliminary research on *Solidago graminifolia* reveals promising potential which could be valued as antioxidant and antimicrobial agent in efficient herbal medicines. Additional studies are requested in order to elucidate the underlying mechanisms of pharmacological properties and to provide safe and active phytopharmaceutical products.

## Figures and Tables

**Figure 1 molecules-24-02666-f001:**
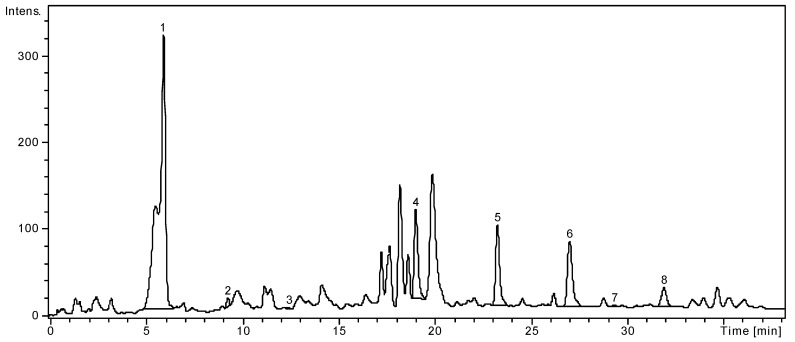
HPLC chromatogram of *Solidago graminifolia* ethanol extract (method *b*), as described in Materials and methods Section 3.5.

**Table 1 molecules-24-02666-t001:** Total phenolic and flavonoid contents (mg/g extract) in *Solidago graminifolia* extracts.

Extract	Yield (mg/g)	Total Phenolic Content (mg GAE/g)	Total Flavonoid Content (mg RE/g)
Petroleum ether extract (PEE)	121.2 ± 1.39	40.67 ± 1.87	98.44 ± 3.81
Chloroform extract (CE)	40.51 ± 1.02	18.74 ± 0.16	45.63 ± 1.06
Ethanol extract (EE)	317.17 ± 3.42	192.69 ± 2.64	151.41 ± 2.44
Methanol extract (ME)	295.4 ± 3.15	179.04 ± 2.55	130.58 ± 2.08
Aqueous extract (AE)	280.01 ± 3.01	166.29 ± 2.08	128.37 ± 1.82

Note: Values are expressed as the mean ± SD (*n* = 3).

**Table 2 molecules-24-02666-t002:** Polyphenolic composition of *Solidago graminifolia* ethanol extract by HPLC (mg/100 g extract).

Polyphenolic Compound	[M − H]^−^ *m*/*z*	Main Daughter Ions	R_T_ ^a,b,c^ ± SD (min)	Polyphenol Content mg/100 g Extract
Caftaric acid	311	148.6, 178.6	3.34 ^b^ ± 0.05	<LOQ
Gentisic acid	153	108.7	3.69 ^b^ ± 0.04	<LOQ
Chlorogenic acid	353	178.7, 190.7	5.6 ^b^ ± 0.05	997.88 ± 7.63
*p*-Coumaric acid	163	118.7	9.18 ^b^ ± 0.08	9.89 ± 0.27
Ferulic acid	193	133.7, 148.7, 177.6	12.8 ^b^ ± 0.10	1.04 ± 0.05
Hyperoside	463	254.9, 270.9, 300.7	19.02 ^b^ ± 0.12	253.19 ± 1.58
Rutin	609	254.9, 270.9, 300.7, 342.8	20.06 ^b^ ± 0.15	<LOQ
Quercitrin	447	178.8, 300.7	23.44 ^b^ ± 0.13	431.59 ± 2.94
Quercetin	301	150.6, 178.6, 272.7	26.8 ^b^ ± 0.11	130.33 ± 1.04
Luteolin	285	150.6, 174.6, 198.6, 240.7	29.64 ^b^ ± 0.19	2.7 ± 0.09
Kaempferol	285	150.6, 256.7	32.48 ^b^ ± 0.07	45.53 ± 3.82
Gallic acid	169	169 *	1.5 ^c^ ± 0.09	3.22 ± 0.11
Protocatechuic acid	153	153 *	2.8 ^c^ ± 0.15	64.08 ± 4.53
Vanillic acid	167	167 *	6.7 ^c^ ± 0.17	2.52 ± 0.08
Syringic acid	197	197 *	8.4 ^c^ ± 0.11	7.19 ± 0.51
Rosmarinic acid	359	160.6, 178.6, 196.7	2.2 ^a^ ± 0.18	19.23 ± 0.88

Notes: Values are expressed as the mean ± SD (*n* = 3); <LOQ—below limit of quantification; *—MS1 (SIM) analysis mode; ^a,b,c^—the HPLC methods as described in Materials and methods Section 3.5.

**Table 3 molecules-24-02666-t003:** Free radical scavenging activity by DPPH and ABTS assays of *Solidago graminifolia* extracts.

Extract	DPPH Scavenging Activity IC_50_ (μg/mL)	ABTS Scavenging Activity (mg TE/g dw)
Ethanol extract (EE)	12.61 ± 0.74	249.55 ± 3.02
Methanol extract (ME)	20.39 ± 0.31	196.81 ± 2.35
Aqueous extract (AE)	28.44 ± 0.59	165.31 ± 2.06

Note: Values are expressed as the mean ± SD (*n* = 3).

**Table 4 molecules-24-02666-t004:** Antibacterial activity of *Solidago graminifolia* extracts.

Extract/Bacterial Strain	*S. aureus*	*P. aeruginosa*	*L. monocytogenes*	*E. coli*	*S. typhimurium*
Ethanol extract MIC (mg/mL)	0.04 ± 0.001	1.56 ± 0.01	1.56 ± 0.02	1.56 ± 0.01	3.12 ± 0.03
Ethanol extract MBC (mg/mL)	0.09 ± 0.002	3.12 ± 0.02	3.12 ± 0.03	3.1 ± 0.02	6.25 ± 0.07
Methanol extract MIC (mg/mL)	0.09 ± 0.001	3.12 ± 0.01	1.56 ± 0.02	3.12 ± 0.01	3.12 ± 0.04
Methanol extract MBC (mg/mL)	0.19 ± 0.001	6.25 ± 0.04	3.12 ± 0.02	6.25 ± 0.05	6.25 ± 0.03
Aqueous extract MIC (mg/mL)	0.09 ± 0.001	3.12 ± 0.01	3.12 ± 0.03	3.12 ± 0.01	3.12 ± 0.04
Aqueous extract MBC (mg/mL)	0.19 ± 0.001	6.25 ± 0.04	6.25 ± 0.05	6.25 ± 0.05	6.25 ± 0.03
Gentamycin MIC (μg/mL)	0.03 ± 0.001	1.2 ± 0.02	0.07 ± 0.001	1.2 ± 0.01	2.4 ± 0.03
Gentamycin MBC (μg/mL)	0.07 ± 0.002	2.4 ± 0.04	0.15 ± 0.01	2.4 ± 0.05	4.8 ± 0.07

Note: Values are expressed as the mean ± SD (*n* = 3).

**Table 5 molecules-24-02666-t005:** Antifungal activity of *Solidago graminifolia* extracts.

Extract/Bacterial Strain	*Aspergillus flavus*	*Aspergillus niger*	*Candida albicans*	*Candida parapsilosis*	*Penicillium fumiculosum*
Ethanol extract MIC (mg/mL)	0.05 ± 0.008	0.05 ± 0.002	0.01 ± 0.006	0.01 ± 0.003	0.05 ± 0.004
Ethanol extract MFC (mg/mL)	0.1 ± 0.002	0.1 ± 0.003	0.02 ± 0.003	0.02 ± 0.002	0.1 ± 0.003
Methanol extract MIC (mg/mL)	0.05 ± 0.006	0.1 ± 0.005	0.02 ± 0.007	0.01 ± 0.004	0.1 ± 0.006
Methanol extract MFC (mg/mL)	0.1 ± 0.002	0.2 ± 0.006	0.05 ± 0.004	0.02 ± 0.006	0.2 ± 0.005
Aqueous extract MIC (mg/mL)	0.1 ± 0.005	0.1 ± 0.004	0.05 ± 0.008	0.05 ± 0.006	0.1 ± 0.007
Aqueous extract MFC (mg/mL)	0.2 ± 0.006	0.2 ± 0.008	0.1 ± 0.006	0.1 ± 0.007	0.2 ± 0.006
Gentamycin MIC (μg/mL)	0.15 ± 0.03	0.15 ± 0.03	0.1 ± 0.02	0.1 ± 0.02	0.15 ± 0.03
Gentamycin MFC (μg/mL)	0.3 ± 0.05	0.3 ± 0.06	0.2 ± 0.04	0.2 ± 0.04	0.3 ± 0.06

Note: Values are expressed as the mean ± SD (*n* = 3).

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
