# Peer review of "Solidago graminifolia L. Salisb. (Asteraceae) as a Valuable Source of Bioactive Polyphenols: HPLC Profile, In Vitro Antioxidant and Antimicrobial Potential"

_molecules, 2019, doi:10.3390/molecules24142666_

Round 1
Reviewer 1 Report
This paper deals with the characterization of the polyphenolic profile, antioxidant and antimicrobial activity of Solidago sp. Authors have recently published many papers on the same topic addressing the characterization of other plants. In my opinion, the paper is interesting and well-written although they should make an effort to point out what is new here and why this paper deserves to be published.
A chromatogram showing the richness and complexity of the extracts regarding to phenolic species will be welcome.
When we compare the total polyphenolic index with the antiradical activity data seems to be uncorrelated, meaning that ethanol, methanol and water extracts slightly differ in TPC while differences among ABTS or DPPH values are more dramatic. Which is the reason of such discrepancies? This issue should be commented in the text.
Table 3. The meaning of the row of trolox is unclear.
The comparison of authors’ results with those previously published in then literature is quite long. I suggest this part (in section 2) could be shortened substantially. Furthermore, the number of references for a regular research paper is excessive.
Author Response
We thank Reviewer for the thoughtful comments. The suggestions have been implemented in the manuscript and detailed in the attached file.

Reviewer 2 Report
The authors have submitted a manuscript in which they analyze the content and the profile of polyphenols of a poorly explored medicinal plant (Solidago graminifolia L. Salisb.), studying also its biological properties, in particular the antibacterial activity against Staphylococcus aureus and important antifungal effect against Candida albicans and C. parapsilosis.
The topic of this paper is suitable for the scope of the Journal. The novelty is due to the use of a poorly studied plant. In this regard, in abstract it is better to us poorly explored rather than unexplored: e.g. there is an article analyzing the characteristics of essential oil extracted from this variety that can be cited (https://doi.org/10.1002/ffj.2730090514)
Lines 116-126: I suggest a better discussion of the effect of the solvent used, that is not only linked to the polarity (as done in 10.3303/CET1439078)
Lines 127-130: Are the extraction condition used in the cited article the same of your work? If not, you should point out the differences
Lines 200-213 the description of the two methods for the determination of the antioxidant activity is very well known in literature and, in particular, for the readers of this Journal. I suggest eliminating this part. On the contrary it can be interesting to point out the results between these two methods and so the activity of your extracts to the two radical reagents used. In order to do this I suggest using Trolox equivalents (TEs)/g of extract for both analyses. IC50 depends to the liquid to solid ratio used and so is difficult to be immediately compared by readers. E.g. in lines 236-239 and 245-247 you compare the IC50 values obtained in other studies, but without the information of the liquid to solid ratio it is impossible to understand the differences.
Perhaps lines 131-135, 189-191, 259-261, 280-282 are worthy of a separate paragraph. Please consider this possibility, even if it is not a problem to leave it that way.
In all tables use the same number of significant digits.
In the last sentence of the Conclusions section, you write that taking into account your results, Solidago graminifolia L. Salisb.has a great potential in herbal medicine, but you did not mention in the text that this plant is edible. If so, it could also be interesting to use it in food supplements.
Other minor remarks:
Revise English grammar, in particular the use of the article “the”
e.g. Line 65 and 97 not the flavonoids, but only flavonoids
Line 70 not the North but only North
Line 108 not the total but only Total
And so on
Line 66 I do not understand the use of “must”
Line 79 revise English
Line 347-350 revise English
Line 360-361 revise English
I suggest eliminating line 388
I suggest eliminating paragraph 3.9
Author Response

(The authors gave the same response as above.)
